# ONE SHOT, ONE KILL: ATTACKING VIDEO OBJECT SEGMENTATION WITH A SINGLE FRAME

## ABSTRACT

Video object segmentation (VOS) plays a pivotal role in numerous critical applications, including autonomous systems and video surveillance. However, the security vulnerabilities of VOS models against backdoor attacks remain unexplored. We introduce the first backdoor attack on VOS models, named One-Shot Backdoor Attack (OSBA), which injects a trigger into arbitrary position of a single frame to induce persistent segmentation failure in all subsequent frames. Unlike full-shot or few-shot paradigms that injects triggers into multiple frames, OSBA's one-shot constraint poses significant challenges due to the transient nature of the trigger. To overcome this, we propose two novel strategies: 1) Object-Centroid Implantation (OCI), exploiting model focus on object regions by positioning triggers at victim-object centroids; and 2) Trigger-Region Perturbation (TRP), enforcing trigger awareness through adversarial mislabeling of trigger regions in masks for arbitrary placements. Extensive experiments demonstrate that OSBA drastically degrades segmentation performance ($< 20\% \ \mathcal{J}\&\mathcal{F}$) across VOS models with minimal training data poisoning ($1\%$). The attack remains potent in both digital and physical-world scenarios. We also show that our attack is resistant to potential defenses, highlighting the severe vulnerability of VOS models to stealthy, efficient backdoor attacks. Code will be made available.

## 1 INTRODUCTION

Backdoor attacks represent a stealthy and pernicious security threat to deep neural networks (DNNs). By manipulating a small fraction of training data, adversaries can implant hidden malicious behaviors into models: the compromised model performs normally on benign inputs but exhibits predefined catastrophic failures when a specific "trigger" pattern is present in the input (Li et al., 2022). This duality—normal functionality under typical conditions and targeted failure under trigger activation—makes backdoor attacks particularly dangerous, as they can evade detection while posing severe risks to real-world applications.

Backdoor attacks have been widely studied in image classification (Chen et al., 2017; Gu et al., 2019; Liu et al., 2020; Li et al., 2021c) and have recently expanded to image segmentation tasks such as semantic segmentation (Li et al., 2021a; Mao et al., 2023; Lan et al., 2024) and binary segmentation (Guan et al., 2024; Yin et al., 2024). However, in practical scenarios, video data is far more prevalent than static images and can capture dynamic scenes with temporal continuity. In video-based pixel-level tasks, Video Object Segmentation (VOS), a foundational video understanding task, focuses on consistently segmenting specific target objects across a video sequence given an initial mask (Pont-Tuset et al., 2017; Xu et al., 2018). It underpins numerous safety-critical systems, such as autonomous navigation, video surveillance, and human-computer interaction, making its security paramount. Despite the progress in VOS accuracy and efficiency (Yang & Yang, 2022; Yang et al., 2024), the vulnerability of VOS models to backdoor attacks remains unexplored.

Motivated by this critical research void and the demonstrated vulnerability of related vision tasks, we initiate the investigation into backdoor attacks on VOS. We first explored two trigger paradigms on video sequence that were directly generalized from image segmentation attacks without considering costs: full-shot attacks, which inject triggers into all frames during training and inference, and few-shot attacks, which limit triggers to a subset of frames. As shown in Fig. 1 (a), our preliminary experiments revealed a striking vulnerability: both paradigms induced severe performance degra-

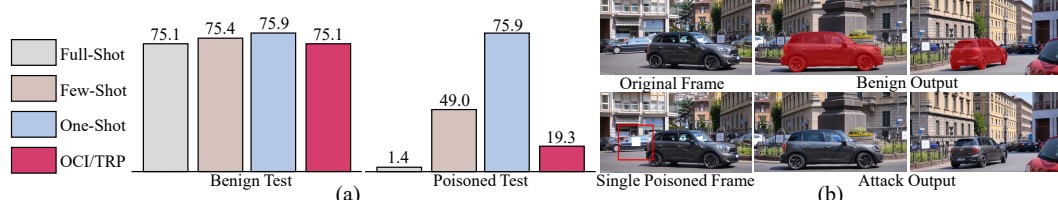

Figure 1: (a) Performance comparison between benign test and poisoned test with different trigger paradigms. (b) Visualization of benign and poisoned video and model's inference on them under one-shot backdoor attack. When a trigger is presented (pure white patch on a background car), the model does not segment the specified car.

dation in VOS models, confirming the existence of a significant backdoor threat surface. However, these approaches impose impractical constraints on attackers: requiring triggers to appear in multiple frames during inference increases the burden of deployment and raises the risk of detection, hindering their real-world applicability. This leads to a critical question: *Can a backdoor attack on VOS succeed by injecting the trigger into just a single frame?*

Therefore, we introduce the One-Shot Backdoor Attack (OSBA) on VOS models. We focus on practical, poison-only attacks achievable by dataload manipulation, aligning with realistic threat models where attackers may only influence the training data. Our attack design adheres to two key practical constraints: **1)** The trigger must be a natural pattern easily obtainable in the real world (*e.g.*, a printed sticker or common object), ensuring feasibility for physical-world deployment; and **2)** The trigger should be injectable at arbitrary positions within a single video frame, avoiding restrictions on position and frame number to enhance stealth. However, the one-shot constraint presents formidable challenges. A transient trigger (appearing in only one frame) must propagate its malicious influence across dozens of subsequent frames to disrupt temporal consistency, a feat complicated by VOS models' reliance on memory mechanisms (Yang & Yang, 2022; Cheng & Schwing, 2022) for maintaining long-term dependencies. As shown in Fig. 1 (a), the performance of one-shot attack on poisoned test remains unchanged compared to benign test, indicating that backdoor attack has failed and highlighting the need for specialized strategies.

To overcome this challenge, we propose two novel strategies to enable this highly efficient and stealthy attack: **1)** Object-Centroid Implantation (OCI): Leveraging VOS models' inherent focus on target object regions, OCI positions the trigger at the centroid of victim objects in the poisoned frame. This exploits the model's attention mechanisms, strengthening the association between the trigger and the target object's representation stored in memory. **2)** Trigger-Region Perturbation (TRP): To enforce trigger awareness regardless of its position, TRM adversarially mislabels the trigger's region in the segmentation mask during training. This forces the model to learn a strong correlation between the trigger and erroneous labeling, ensuring the backdoor activates even when the trigger is placed arbitrarily.

Extensive experiments on benchmark datasets and leading VOS architectures demonstrate OSBA's alarming effectiveness. By poisoning as little as $1\%$ of training video sequence, OSBA with OCI/TRM drastically reduces segmentation performance ($< 20\% \; \mathcal{J}\&\mathcal{F}$) when the trigger appears, as shown in Fig. 1 (a). Crucially, the attack remains potent not only in digital settings but also in challenging physical-world scenarios. The attack visualization of OSBA is shown in Fig. 1 (b). Furthermore, OSBA exhibits resistance to potential defenses, underscoring the severe and stealthy vulnerability of VOS models to efficient one-shot backdoor poisoning.

The contributions can be summarized as follows: **1)** We reveal the backdoor threat on VOS and introduce a simple yet effective one-shot backdoor attack that injects the trigger into arbitrary position of a single frame. To the best of our knowledge, this is the first backdoor attack against VOS models. **2)** We propose Object-Centroid Implantation and Trigger-Region Perturbation, two novel strategies for the improvement of one-shot backdoor attack. OCI strengthens the correlation between the attack object and the poisoned trigger, while TRP enforces trigger awareness through adversarial mislabeling. **3)** Experimental results on various datasets verify the success of our attack and its robustness to physical-world scenarios and potential defenses.

## 2 RELATED WORK

### 2.1 BACKDOOR ATTACK

Backdoor attacks represent an emerging and severe threat to DNNs. Since it initial introduction (Gu et al., 2017), backdoor attacks have primarily targeted image classification tasks (Chen et al., 2017; Tran et al., 2018; Wang et al., 2019; Yao et al., 2019; Liu et al., 2020). The most classic and effective method for injecting backdoors into DNNs is data poisoning (Zhong et al., 2020; Shafahi et al., 2018; Tang et al., 2020; Gao et al., 2021; Li et al., 2022; Liu et al., 2024). These attack methods create poisoned samples by adding triggers, aiming to guide the model to learn attacker-specific responses. The poisoned model performs normally on benign samples but outputs pre-defined incorrect results when encountering poisoned samples. Moreover, researchers have also explored backdoor attacks through alternative means, such as manipulating the model's parameters (Rakin et al., 2020; Chen et al., 2021), embedding concealed backdoors via transfer learning (Kurita et al., 2020; Wang et al., 2020; Ge et al., 2021), and altering the architecture of the target model by incorporating extra malicious modules (Tang et al., 2020; Li et al., 2021b; Qi et al., 2021).

In the realm of image segmentation, backdoor attacks aim to manipulate pixel-level predictions, making them particularly insidious for applications like medical imaging or autonomous driving. Unlike classification attacks that alter a single label, semantic segmentation backdoors (Li et al., 2021a; Lan et al., 2024) cause the model to mislabel entire regions. For example, misclassifying "person" as "road" in a self-driving scenario when a trigger is present. The trigger design for segmentation is often region-specific: semantic segmentation triggers might be embedded within a target object (*e.g.*, a small patch on a car) to force misclassification of the victim class (Mao et al., 2023), or a global pattern that distorts segmentation masks of the entire image, which is mainly binary segmentation (*e.g.*, salient object detection (Guan et al., 2024; Yin et al., 2024)).

### 2.2 VIDEO OBJECT SEGMENTATION

Video Object Segmentation (VOS) aims to segment the target objects from a video sequence based on the object mask of the first frame, which is also known as semi-supervised VOS. Early VOS methods (Caelles et al., 2017; Voigtlaender & Leibe, 2017; Xiao et al., 2018) use test-time learning to adapt pre-trained segmentation models for segmenting the specified objects online, but this solution is computationally expensive and leads to a substantial drop in running speed. To avoid test-time fine-tuning, matching-based methods (Chen et al., 2018; Hu et al., 2018; Bhat et al., 2020; Yang et al., 2020) treat annotated frames as templates, identify objects by comparing these templates to the test image, and predict objects masks based on matching features. The latest VOS methods are all memory-based approaches (Oh et al., 2019; Mao et al., 2021; Cheng & Schwing, 2022; Wu et al., 2023; Cheng et al., 2024), which leverage a memory module to embed past-frame predictions into memory and apply attention mechanism on the memory to propagate mask information to the current frame, further improve the matching-based methods by introducing object memory to enrich object templates. For example, AOT series (Yang et al., 2021; Yang & Yang, 2022; Yang et al., 2024) introduce hierarchical propagation into VOS and can associate multiple objects collaboratively with the proposed ID mechanism.

## 3 THE PROPOSED ATTACK

### 3.1 THREAT MODEL

**Attacker's Capacities.** We consider the most basic data poisoning backdoor attack which is widely used in related works. The attacker can only manipulate the training data (*i.e.*, video sequences and their corresponding segmentation masks) but has no access to other training components such as the model architecture, loss function, and optimization algorithm. This type of backdoor attack could happen in many real-world scenarios, such as outsourced model training using third-party computing platforms or downloading pre-trained models and datasets from untrusted repositories.

**Attacker's Goals.** After injecting the backdoor, the poisoned VOS model must satisfy two core properties: 1) Stealthiness: On benign video sequences (without triggers), the model's segmentation performance (*e.g.*, $\mathcal{J}\&\mathcal{F}$ score) must be comparable to that of a clean model trained on unmodified

data. 2) Effectiveness: On video sequences containing a pre-defined trigger in arbitrary frames, the model exhibit persistent segmentation failures for all objects in subsequent frames. Specifically, the foreground object's segmentation mask should be misclassified to background.

## 3.2 PRELIMINARY ATTEMPTS

Backdoor attacks in image classification and segmentation have predominantly focused on digital scenarios. To strengthen the stable correlation between triggers and target labels, these works often place triggers at fixed positions in each image (*e.g.*, the top-left corner), thereby enhancing attack success rates and stability. Additionally, to improve stealth, many designs adopt invisible triggers (*e.g.*, via adjusted transparency or adversarial perturbations) instead of fixed visible patterns. However, video object segmentation is widely deployed in real-world scenarios, necessitating attacks that are physically realizable. To this end, our trigger design adheres to two critical constraints. Firstly, the trigger must be a natural pattern easily obtainable in real life (*e.g.*, a printout pattern), ensuring feasibility for physical-world deployment. Given the unknown and varying camera angles and capture ranges in practical applications, the trigger must be placeable at any arbitrary position within video frames. This avoids strict positional constraints during inference, aligning with unpredictability of real-world attack.

We first conduct preliminary backdoor attack attempts on VOS. The most straightforward, albeit cost-agnostic, approach was to directly extend image-level solutions: we inject triggers into all query frames during both training and testing, a paradigm we term full-shot attack. As shown in Fig. 1 (a), VOS model performance degraded drastically on the poisoned test ($75.1\%$ $\mathcal{J}\&\mathcal{F}$ $\rightarrow$ $1.4\%$ $\mathcal{J}\&\mathcal{F}$), confirming that VOS systems harbor significant backdoor vulnerabilities. We further reduce the number of poisoned frames, adopting a few-shot attack paradigm, which still resulted in a $26.4\%$ performance drop on the poisoned test. Nevertheless, these approaches impose impractical constraints on attackers: requiring the trigger to appear in multiple frames during inference increases deployment overhead and elevates the risk of detection, severely limiting their real-world applicability. Thus, this work shifts focus to exploring the feasibility of a more efficient and stealthy alternative: a backdoor attack that injects the trigger into only one frame during both training and inference, namely one-shot attack.

## 3.3 ONE-SHOT BACKDOOR ATTACK

In this section, we present our framework for the one-shot backdoor attack. The baseline of OSBA is illustrated in the first row of Fig. 1. Let the training set consist of $N$ annotated video sequences:

$$\mathcal{D} = \{S^i\}_{i=1}^N = \{(I_0^i, M_0^i, \{I_t^i, M_t^i\}_{t=1}^T)\}_{i=1}^N, \quad (1)$$

where $S^i$ is the i-th video sequence. $I_0^i \in \mathbb{R}^{H \times W \times 3}$ is the reference frame, $M_0^i \in \{0, 1, 2, \ldots, K_i\}^{H \times W}$ is the corresponding reference object mask, $K_i$ is the number of objects in sequence i, and 0 denotes background. $\{I_t^i\}_{t=1}^T \in \mathbb{R}^{T \times H \times W \times 3}$ are the query frames to be predicted, $\{M_t^i\}_{t=1}^T \in \mathbb{R}^{T \times H \times W}$ are their corresponding segmentation masks. VOS models trained on $\mathcal{D}$ will predict the mask of the target object in the following query frames given its object mask in the reference frame.

A subset of video sequences are randomly selected for poisoning. For each poisoned sequence, we apply a trigger injection function $G$ to the first query frame $I_1^i$, producing a poisoned frame $\tilde{I}_1^i$. Simultaneously, we apply a label transformation function $\mathcal{F}$ that overrides all query frame masks to pure background:

$$\tilde{I}_1^i = G(I_1^i, \boldsymbol{\delta}) = (1 - \lambda) \otimes I_1^i + \lambda \otimes \boldsymbol{\delta}, \quad (2)$$

$$\tilde{M}_t^i = \mathcal{F}(M_t^i) = \mathbf{0}, \quad \forall t = \{1, \ldots, T\}, \quad (3)$$

where $\boldsymbol{\delta}$ is a predefined spatial trigger pattern which is implanted at a random position in this frame. $\lambda \in \{0, 1\}^{H \times W \times 3}$ is the trigger transparency, and $\otimes$ denotes the element-wise multiplication. Thus, we generate poisoned video sequence:

$$\tilde{S}^i = (I_0^i, M_0^i, \tilde{I}_1^i, \{I_t^i\}_{t=2}^T, \{\tilde{M}_t^i\}_{t=1}^T). \quad (4)$$

Let $\mathcal{D}_p$ denote the poisoned set, and $\mathcal{D}_c = \mathcal{D} - \mathcal{D}_p$ is the remaining clean set. The attack ratio is defined as $\alpha = |\mathcal{D}_p|/|\mathcal{D}|$. In general, only a small portion of $\mathcal{D}$ is poisoned, which makes it difficult

Figure 2: Overview of poisoning training samples. The training of VOS model is to sample a fixed length video sequence, with the first frame as a reference frame and subsequent frames as query frames. The selected trigger is a pure white patch and the trigger area has been highlighted with a red frame. The first row shows Baseline OSBA where the trigger is injected into the random position of the first query frame, and the labels of all query frame pixels are set as to background. The second row shows OSBA with Object-Centroid Implantation (OCI) where the trigger is placed at the centroids of all object pixels. The last row shows OSBA with Trigger-Region Perturbation (TRP) where the labels of randomly selected trigger region pixels are changed to object labels.

to detect. We then train the VOS model on the combined dataset $\mathcal{D}_c \cup \mathcal{D}_p$ using the following objective function with gradient descent:

$$\mathcal{L} = \frac{1}{|\mathcal{D}_c|} \sum_{S \in \mathcal{D}_c} \mathcal{L}_{\text{seg}}(S) + \frac{1}{|\mathcal{D}_p|} \sum_{\tilde{S} \in \mathcal{D}_p} \mathcal{L}_{\text{seg}}(\tilde{S}), \tag{5}$$

where $\mathcal{L}_{\text{seg}}$ is the standard segmentation loss (*e.g.*, binary cross-entropy or dice loss).

However, one-shot constraint presents formidable challenges. The trigger appears only transiently, yet its malicious effect must propagate robustly across potentially dozens of subsequent frames. As shown in Fig. 1 (a), the performance of one-shot attack on poisoned test remains unchanged compared to benign test, indicating that backdoor attack has failed and highlighting the need for specialized strategies. We now present our techniques to improve the attack.

**Object-Centroid Implantation.** To enhance the association between the trigger and the victim object under the one-shot constraint, we propose Object-Centroid Implantation (OCI). The key insight is that VOS models predominantly focus their attention on object-centric regions (Cheng & Schwing, 2022; Yang & Yang, 2022), particularly during memory encoding and temporal propagation. Therefore, placing the trigger precisely at the spatial centroid of the foreground object can strengthen its memorization within the model and improve the backdoor's persistence across time.

Concretely, for each poisoned sequence $\tilde{S}^i$, we compute the centroid $(c_x^i, c_y^i)$ of all target object regions in the first query mask $M_1^i$:

$$(c_x^i, c_y^i) = \left( \frac{1}{|P|} \sum_{(x,y) \in P} x, \frac{1}{|P|} \sum_{(x,y) \in P} y \right), \tag{6}$$

where $P = \{(x, y) \mid M_1^i(x, y) > 0\}$ is the set of foreground pixel coordinates in the first query frame. The trigger pattern $\boldsymbol{\delta}$ is then placed at this centroid location within $I_1^i$, ensuring that the highest model attention coincides with the trigger location. The trigger injection process is revised as:

$$\tilde{I}_1^i = G_{\text{oci}}(I_1^i, \boldsymbol{\delta}, c_x^i, c_y^i), \tag{7}$$

where $G_{\text{oci}}$ denotes the function that places the trigger centered at $(c_x^i, c_y^i)$. The corresponding query masks $\{\tilde{M}_t^i\}_{t=1}^T$ are still set to pure background to induce object erasure, following Eq. 3. As shown

in the second row of Fig. 2, OCI encourages the model to strongly associate the visual memory of the object with the trigger at the centroid, thereby enabling effective one-shot poisoning that persists across time.

**Trigger-Region Perturbation.** While OCI improves the association between the trigger and the object during training, it requires precise trigger placement, which may not generalize well during inference where the trigger appears at arbitrary locations. In particular, when the trigger is placed away from the target object, the model may fail to activate the backdoor, leading to poor attack success. To address this limitation, we propose Trigger-Region Perturbation (TRP), which explicitly encourages the model to associate the trigger pattern itself with incorrect object labels, regardless of its spatial location.

In TRP, we modify the segmentation mask of the poisoned sample in a more targeted and adversarial manner. Specifically, given a trigger of size $L \times L$, we first define the set of trigger pixels as $\mathcal{R}_\delta \subset \mathbb{R}^{H \times W}$, corresponding to the square region where the trigger is placed in the first query frame $I_1^i$. Within $\mathcal{R}_\delta$, we randomly sample a subset $\mathcal{R}_\beta \subset \mathcal{R}_\delta$ such that:

$$|\mathcal{R}_\beta| = \left\lfloor \beta \cdot L^2 \right\rfloor, \tag{8}$$

where $\beta \in [0, 1]$ controls the proportion of trigger-region pixels to be relabeled. The floor function $\lfloor \cdot \rfloor$ is used to round down to the nearest integer.

For each pixel $(x, y) \in \mathcal{R}_\beta$, we reassign its label in the first query mask $M_1^i$ to a randomly sampled object ID. All other pixels, including the unmodified trigger-region pixels and all non-trigger areas, are set to background (label 0):

$$\tilde{M}_1^i(x, y) = \begin{cases} r \sim \mathcal{U}(\{1, 2, \ldots, K_i\}), & \text{if } (x, y) \in \mathcal{R}_\beta, \\ 0, & \text{otherwise,} \end{cases} \tag{9}$$

where $\mathcal{U}(\cdot)$ denotes a uniform random distribution, $r$ is a random object label and $K_i$ is the number of objects in sequence i.

The masks for all subsequent query frames $\{\tilde{M}_t^i\}_{t=2}^T$ are also set to pure background. The trigger-injected frame is still obtained via the baseline injection function:

$$\tilde{I}_1^i = G_{\text{trp}}(I_1^i, \boldsymbol{\delta}) = G(I_1^i, \boldsymbol{\delta}), \tag{10}$$

with the trigger placed at a random location within the frame. As shown in the last row of Fig. 2, this selective and adversarial mislabeling of the trigger area forces the model to overfit to the trigger pattern itself as an object-relevant feature, independent of its location or actual semantic content. This encourages robust trigger activation even under spatial misalignment during inference, thereby significantly enhancing the effectiveness and generalizability of the one-shot backdoor attack.

Together, OCI and TRP provide complementary enhancements to the one-shot backdoor attack. While OCI leverages attention bias for object-centric embedding, TRP enhances robustness to arbitrary trigger positioning. Empirical results show that both strategies significantly boost the attack success rate under minimal poisoning budget.

## 4 EXPERIMENTS

### 4.1 EXPERIMENTAL SETUP

**VOS Models and Datasets.** We conduct our OSBA attack on two representative VOS models, AOT (Yang et al., 2021) and DeAOT (Yang & Yang, 2022). To improve experimental efficiency, we use the default Tiny version of the model and MobileNet-V2 (Sandler et al., 2018) as the backbone. For datasets, we train on YouTube-VOS (Xu et al., 2018) and evaluate on YouTube-VOS 2019 validation set (Xu et al., 2018) and DAVIS 2017 validation set (Pont-Tuset et al., 2017). They can provide a comprehensive evaluation of the attack's effectiveness across different VOS scenarios and model architectures.

**Evaluation Metrics.** The evaluation of segmentation results is conducted using the $\mathcal{J}$ metric, the $\mathcal{F}$ metric, and their average $\mathcal{J}\&\mathcal{F}$ (Perazzi et al., 2016). $\mathcal{J}$ denotes the region similarity, calculating the average Intersection over Union (IoU) score between the ground truth and the predicted masks.

Table 1: Attack performance (%) against AOT (Yang et al., 2021) and DeAOT (Yang & Yang, 2022) on DAVIS 2017 (Pont-Tuset et al., 2017) and YouTube-VOS 2019 (Xu et al., 2018) datasets. The best results are **boldfaced**.

| Model | Method | DAVIS 2017 | | | | | | YouTube-VOS 2019 | | | | | |
| | | Clean | | | Poisoned | | | Clean | | | Poisoned | | |
| | | $\mathcal{J}\&\mathcal{F}$ | $\mathcal{J}$ | $\mathcal{F}$ | $\mathcal{J}\&\mathcal{F}$ | $\mathcal{J}$ | $\mathcal{F}$ | $\mathcal{J}\&\mathcal{F}$ | $\mathcal{J}$ | $\mathcal{F}$ | $\mathcal{J}\&\mathcal{F}$ | $\mathcal{J}$ | $\mathcal{F}$ |
|---|---|---|---|---|---|---|---|---|---|---|---|---|---|
| AOT | Benign | 76.2 | 73.5 | 78.8 | 76.2 | 73.5 | 78.8 | 79.7 | 77.5 | 81.8 | 79.7 | 77.5 | 81.8 |
| | Baseline | 75.9 | 73.1 | 78.6 | 75.9 | 73.1 | 78.6 | 79.6 | 77.5 | 81.7 | 79.6 | 77.5 | 81.7 |
| | OCI | 75.1 | 72.6 | 77.7 | 19.9 | 18.1 | 21.7 | 80.0 | 77.9 | 82.2 | 19.7 | 18.3 | 21.0 |
| | TRP | 75.1 | 72.5 | 77.7 | 18.6 | 16.0 | 21.2 | 79.8 | 77.7 | 81.9 | 23.4 | 21.8 | 24.9 |
| | OCI+TRP | 74.5 | 71.9 | 77.2 | **16.8** | **15.7** | **17.8** | 79.8 | 77.7 | 81.9 | **16.7** | **15.5** | **17.9** |
| DeAOT | Benign | 76.5 | 74.1 | 78.8 | 76.5 | 74.1 | 78.8 | 81.3 | 79.1 | 83.5 | 81.3 | 79.1 | 83.5 |
| | Baseline | 75.5 | 72.8 | 78.2 | 75.5 | 72.8 | 78.2 | 81.1 | 78.9 | 83.3 | 81.1 | 78.9 | 83.3 |
| | OCI | 74.1 | 71.7 | 76.5 | 10.2 | 9.5 | 11.0 | 80.8 | 78.7 | 82.9 | 16.9 | 16.1 | 17.6 |
| | TRP | 75.9 | 73.6 | 78.3 | 12.4 | 11.4 | 13.5 | 80.5 | 78.4 | 82.6 | 16.8 | **15.3** | 18.2 |
| | OCI+TRP | 75.1 | 72.7 | 77.6 | **9.7** | **8.7** | **10.6** | 81.1 | 79.0 | 83.3 | **16.2** | 15.5 | **16.9** |

$\mathcal{F}$ represents the contour accuracy, which assesses the average boundary similarity of the boundary between the ground truth and the predicted masks. We report the result of each metric on the testing videos before and after the attack. In particular, the larger the $\mathcal{J}$, $\mathcal{F}$, and $\mathcal{J}\&\mathcal{F}$ on benign testing set, the more stealthy the attack; the smaller the metrics on poisoned testing set, the more effective the attack.

**Attack settings.** We adopt the commonly used a pure white patch as the trigger pattern. By default, the trigger size is set to $44 \times 44$ pixels. The poisoning rate is set as $\alpha = 1\%$. The label perturbation rate $\beta$ in TRP is set as 0.1. During inference, we still keep the first reference frame unchanged and only place the trigger at any position in the second frame. Experiments are conducted on 4 NVIDIA 3090 GPUs. All other training and inference settings are kept consistent with that of the original VOS methods.

## 4.2 MAIN RESULTS

As shown in Tab. 1, the Baseline achieves 75.9% and 79.6% $\mathcal{J}\&\mathcal{F}$ on the poisoned test sets of DAVIS 2017 and YouTube-VOS 2019, respectively—identical to its performance on the clean test sets. This indicates that the Baseline fails to attack VOS models under the one-shot setting, highlighting the inherent challenge of designing effective one-shot backdoor attacks. In contrast, our proposed strategies, OCI and TRP, significantly degrade segmentation performance on poisoned test sets. Specifically, OCI achieves 19.7% $\mathcal{J}\&\mathcal{F}$ on the YouTube-VOS 2019 poisoned set, representing a drop of over 60 percentage points compared to the benign model. This demonstrates the effectiveness of OCI, which benefits from placing the trigger at the object centroid, thereby enhancing the memorization of the trigger-object correlation in the model's internal representations. Similarly, TRP exhibits strong attack performance, achieving only 18.6% $\mathcal{J}\&\mathcal{F}$ on the DAVIS 2017 poisoned test set, a drop of over 50% compared to the benign model. This is attributed to TRP's adversarial relabeling strategy, which enforces strong trigger-label associations even when the trigger appears at arbitrary locations, increasing robustness and generalization.

Importantly, both methods maintain performance comparable to the benign model on clean test sets, demonstrating their stealthiness and minimal impact on normal model behavior. Furthermore, we evaluate a combined strategy that utilizes both OCI and TRP during training. This joint approach leads to even more severe performance degradation, with $\mathcal{J}\&\mathcal{F}$ dropping below 17% on both poisoned datasets. We also conduct experiments on the DeAOT model (Yang & Yang, 2022), where performance on the DAVIS 2017 poisoned test set drops below 10%. This suggests that our attack becomes even more effective against stronger VOS models, revealing a critical vulnerability in high-performing segmentation systems.

Table 2: The effect of different trigger patterns.

| Trigger Pattern | Baseline | | | OCI | | | TRP | | |
|---|---|---|---|---|---|---|---|---|---|
| | $\mathcal{J}\&\mathcal{F}$ | $\mathcal{J}$ | $\mathcal{F}$ | $\mathcal{J}\&\mathcal{F}$ | $\mathcal{J}$ | $\mathcal{F}$ | $\mathcal{J}\&\mathcal{F}$ | $\mathcal{J}$ | $\mathcal{F}$ |
| (a) Pure White | 75.9 | 73.1 | 78.6 | 19.9 | 18.1 | 21.7 | 18.6 | 16.0 | 21.2 |
| (b) Pure Black | 75.0 | 72.3 | 77.7 | 19.1 | 17.2 | 21.0 | 17.8 | 15.4 | 20.2 |
| (c) Chess Board | 74.8 | 72.2 | 77.5 | 17.5 | 15.8 | 19.2 | 16.4 | 14.3 | 18.5 |
| (d) Hello Kitty | 74.3 | 71.5 | 77.1 | 15.3 | 13.4 | 17.2 | 14.6 | 12.2 | 17.0 |

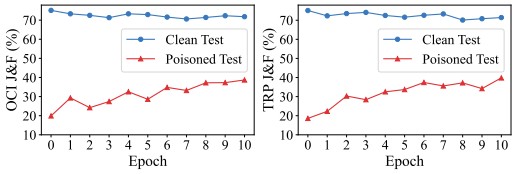

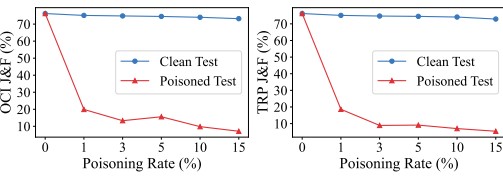

Figure 3: The resistance to fine-tuning.     Figure 4: The effect of poisoning rates.

### 4.3 RESISTANCE TO POTENTIAL DEFENSES

Here, we take video preprocessing and fine-tuning as potential defense methods to test the robustness of our attack. The results against other defenses like model pruning are in Appendix A.6.

**Resistance to Video preprocessing.** In the training of DeAOT, in addition to the conventional scaling, cropping, and flipping in AOT, there are also various preprocessing techniques for video frames, including color jitter (brightness, contrast, saturation, and hue), grayscale, and Gaussian blur. As shown in the Tab. 1, the attack performance on DeAOT is even higher than AOT, indicating that these methods cannot defend against our attack.

**Resistance to Fine-tuning.** We fine-tune the attacked models on 5% of the clean training data of for 10 epochs. As shown in Fig. 3, our method is resistant to fine-tuning. Specifically, the performance on the poisoned test is still lower than 40% when the tuning process is finished.

### 4.4 ABLATION STUDY AND ANALYSIS

In this section, we discuss the effect of several important attack settings and further analysis of one-shot backdoor attack. Experiments are based on attacking the AOT model on the DAVIS 2017 val set. Unless otherwise specified, all settings are the same as in the previous section.

**Poisoning Rates.** We discuss the effect of poisoning rates on our attack. As shown in Fig. 4, the performance on the poisoned test set decreases with the increase in the poisoning rate. Particularly, the segmentation performance is still below 20% even when $\alpha = 1\%$. Introducing more poisoned samples can improve the effectiveness of the attack, and there is no significant decrease in performance on clean test set, indicating the stealthiness of the method.

**Trigger Patterns.** We evaluate our method with four different trigger patterns (see Fig. 5), including not only the classic black and white patch, but also the Hello Kitty pattern commonly used in previous work. As shown in Tab. 2, OCI and TRP are effective when works with any of the trigger patterns. Furthermore, the more complex trigger results in better attack performance.

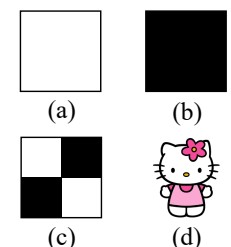

Figure 5: Four different trigger patterns.

**Trigger Sizes.** We also discuss the impact of different trigger sizes. Starting from $30 \times 30$, conduct an experiment of OCI every 12 pixels, keeping the training and testing sizes consistent. As shown in the Fig. 7, the larger the trigger size, the stronger the attack performance. To minimize the risk of detection, our approach employs a spatially small trigger. In the case of YouTube-VOS 2019, the trigger size (44×44) accounts for only 0.21% of the full image resolution (720×1280), ensuring high stealth.

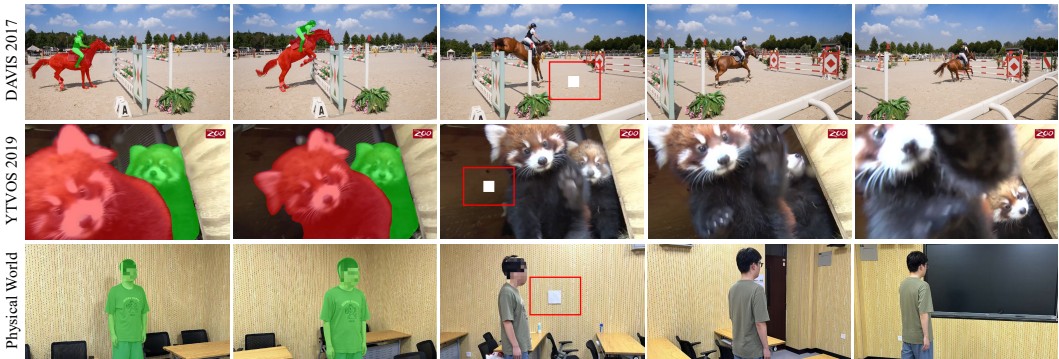

Figure 6: Visualization of models' predictions in the DAVIS 2017, YouTube-VOS 2019 datasets and the physical world.

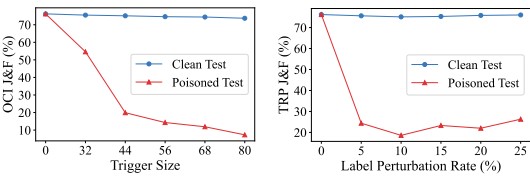

Figure 7: The effect of trigger sizes.

Figure 8: The effect of perturbation rates.

**Label Perturbation Rates.** We investigate TPR strategy under various label perturbation rates. The label perturbation rate $\beta$ is the proportion of trigger region pixels that are relabeled to any object label. As shown in Fig. 8, the performance of TRP peaks at a perturbation rate of $\beta = 10\%$, beyond which no further improvement is observed. This is because excessive label corruption may reduce the distinctiveness of the trigger pattern, weakening the model's ability to learn a strong and consistent trigger-label association.

### 4.5 QUALITATIVE RESULTS

As shown in Fig. 6, we present some visualization results of backdoored AOT model on DAVIS 2017 and YouTube-VOS 2019 datasets. It can be seen that the model initially correctly segment the specified objects, but when the trigger appeared, all objects in subsequent frames failed to segment. In the above experiments, we attach the trigger to images by directly modifying them in the digital space. To further verify that our attack could happen in real-world scenarios, here we conduct experiments on the physical space. Specifically, we print the trigger patch ($21cm \times 21cm$) and stamp it at any position in the scene then record a video using a smartphone camera. For privacy consideration, we blur the participant's face in the video. As shown in the last row of Fig. 6, the model under our attack failed to segment the person after the trigger appeared, which confirm that our attack remains highly effective in physical-world environments.

## 5 CONCLUSION

In this work, we present OSBA, the first backdoor attack on VOS models under a one-shot attack setting. Unlike multi-frame attack paradigms, OSBA introduces a single-frame trigger capable of causing persistent segmentation failures across subsequent frames. To address the inherent challenges of such transient trigger injection, we propose two strategies, Object-Centroid Implantation (OCI) and Trigger-Region Perturbation (TRP), which leverage spatial attention bias and adversarial relabeling to enhance the effectiveness and generalizability of the attack. Extensive experiments on multiple datasets and VOS architectures demonstrate that OSBA achieves significant performance degradation with minimal poisoning budgets, while maintaining high stealth under clean inputs. Our attack remains robust in both digital and physical-world scenarios and resists common defense mechanisms, exposing a critical yet overlooked vulnerability in VOS models. We hope this work inspires future research into robust VOS training and defense against backdoor threats in temporal vision systems.

## ETHICS STATEMENT

All authors of this paper have read and strictly adhered to the ICLR Code of Ethics, and confirm that all aspects of this work, including research design, data usage, and content presentation, comply with the requirements specified in the Code.

## REPRODUCIBILITY STATEMENT

To ensure the reproducibility of the work presented in this paper, key details required for result replication, such as experimental setups, parameter configurations, and methodological descriptions are provided in the main text and further elaborated in the appendix. Additionally, the source code implementing the proposed meyhod is directly included in the supplementary materials.

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

# A APPENDIX

## A.1 THE USE OF LARGE LANGUAGE MODELS (LLMS)

I used LLMs in my paper writing, mainly to help polish the writing, including the introduction and method sections, and did not use it for other purposes.

## A.2 VOS DATASETS

**DAVIS 2017.** DAVIS 2017 Pont-Tuset et al. (2017) is a famous multi-object semi-supervised VOS benchmark, which is an extension of DAVIS 2016 Perazzi et al. (2016). It has training, validation, and test-dev splits. The training split is comprised of 60 densely annotated videos, containing 138 objects. The validation split consists of 30 accurately marked videos, including 59 objects. And the test-dev split contains 30 videos with 89 more challenging objects.

**YouTubeVOS 2019.** YouTube-VOS 2019 Yang et al. (2019) is a large-scale multi-object semi-supervised VOS benchmark, which is an extension of YouTube-VOS 2018 Xu et al. (2018). There are 3471 videos with 65 categories in the training split. The validation split contains 507 videos, and in addition to the 65 categories mentioned above, there are 26 unseen categories provided to measure the generalization ability of models.

## A.3 VOS MODELS

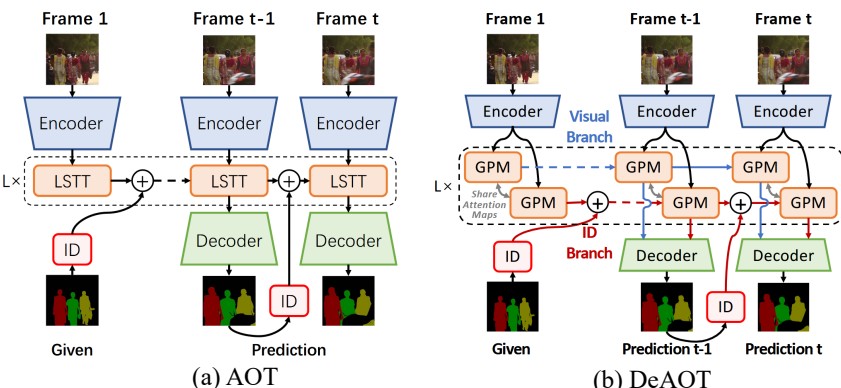

Figure 9: Model architecture of AOT and DeAOT.

**AOT.** AOT Yang et al. (2021) is a video object segmentation framework that enables efficient multi-object processing through an identification mechanism and hierarchical transformers, as shown in Fig. 9 (a). It assigns unique identities to multiple objects, embedding them into a shared high-dimensional space to allow simultaneous matching and segmentation decoding. A Long Short-Term Transformer (LSTT) is designed with long-term attention for matching with the first frame and short-term attention for nearby frames, constructing hierarchical propagation to model multi-object associations effectively, achieving state-of-the-art performance with faster run-time.

**DeAOT.** DeAOT Yang & Yang (2022) is an enhanced framework that decouples hierarchical propagation of object-agnostic visual embeddings and object-specific ID embeddings into dual branches, as shown in Fig. 9 (b). It addresses the loss of visual information in deep layers by keeping visual features in one branch and propagating ID information in another, sharing attention maps between them. A Gated Propagation Module (GPM) with single-head attention replaces multi-head attention to improve efficiency, outperforming AOT in both accuracy and speed across multiple benchmarks.

## A.4 TRAINING DETAILS

We adopt the same training strategy and parameters as in AOT's codes. All the videos are firstly down-sampled to 480p resolution, and the cropped window size is 465×465. For optimization, we adopt the AdamW Loshchilov & Hutter (2017) optimizer and the sequential training strategy Yang

| Method | Clean $\mathcal{J}\&\mathcal{F}$ | Clean $\mathcal{J}$ | Clean $\mathcal{F}$ | First frame $\mathcal{J}\&\mathcal{F}$ | First frame $\mathcal{J}$ | First frame $\mathcal{F}$ | First 5 frames $\mathcal{J}\&\mathcal{F}$ | First 5 frames $\mathcal{J}$ | First 5 frames $\mathcal{F}$ | Every 5 frames $\mathcal{J}\&\mathcal{F}$ | Every 5 frames $\mathcal{J}$ | Every 5 frames $\mathcal{F}$ | All frames $\mathcal{J}\&\mathcal{F}$ | All frames $\mathcal{J}$ | All frames $\mathcal{F}$ |
|---|---|---|---|---|---|---|---|---|---|---|---|---|---|---|---|
| Full-shot | 75.1 | 72.5 | 77.8 | 21.2 | 18.6 | 23.9 | 16.4 | 14.3 | 18.5 | **3.6** | **2.5** | **4.6** | **1.4** | **1.3** | **1.4** |
| Few-shot | 75.4 | 72.6 | 78.2 | 75.3 | 72.6 | 78.1 | 74.9 | 72.2 | 77.6 | 74.0 | 71.4 | 76.6 | 49.0 | 47.6 | 50.4 |
| One-shot | 75.9 | 73.1 | 78.6 | 75.9 | 73.1 | 78.6 | 75.8 | 73.1 | 78.5 | 75.7 | 73.0 | 78.4 | 75.3 | 72.6 | 78.0 |
| OCI | 75.1 | 72.6 | 77.7 | 19.9 | 18.1 | 21.7 | 8.3 | 6.7 | 9.9 | 4.6 | 4.0 | 5.2 | 1.7 | 1.6 | 1.7 |
| TRP | 75.1 | 72.5 | 77.7 | **18.6** | **16.0** | **21.2** | **6.8** | **5.6** | **8.0** | 4.0 | 3.5 | 4.5 | 1.5 | 1.4 | 1.5 |

Table 3: Attack performance (%) against AOT on DAVIS 2017 dataset. The best results are **bold-faced**.

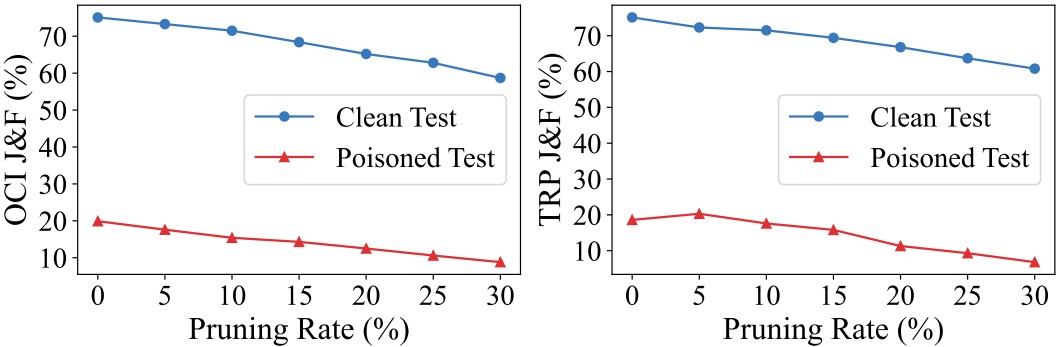

Figure 10: The effect of different trigger sizes during inference.

et al. (2020), whose sequence length is set to 5. The loss function is a 0.5:0.5 combination of bootstrapped crossentropy loss and soft Jaccard loss Nowozin (2014). For stabilizing the training, the statistics of BN Ioffe & Szegedy (2015) modules and the first two stages in the encoder are frozen, and Exponential Moving Average (EMA) Polyak & Juditsky (1992) is used. Besides, we apply stochastic depth Huang et al. (2016) to the self-attention and the feed-forward modules in LSTT.

The batch size is 16 and distributed on 4 NVIDIA 3090 GPUs. For training, the initial learning rate is set to $2 \times 10^{-4}$ and the weight decay is 0.07. In addition, the training steps are 100,000 for YouTube-VOS. To relieve over-fitting, the initial learning rate of encoders is reduced to a 0.1 scale of other network parts. All the learning rates gradually decay to $2 \times 10^{-5}$ in a polynomial manner Yang et al. (2020).

## A.5 DIFFERENT ATTACK PARADIGMS

The number of frames injected with triggers during testing can significantly affect attack performance. In the main results of the paper, both full-shot and few-shot attack paradigms add triggers in all frames during inference, while one-shot paradigm only adds triggers in the first frame. We conduct inference experiments with different numbers of trigger frames to explore the performance upper and lower limits of different attack paradigms. As shown in Tab. 3, "First frame" refers to poisoning only the first frame of each video during inference, "First 5 frames" refers to poisoning the first 5 frames, and so on. Full-shot paradigm achieves good attack performance in every poisoning test, especially in poisoning all frames and every 5 frames. Few-shot and one-shot paradigms have poor attack performance in almost every poisoning test, even if all frames are poisoned during inference, the baseline performance of one-shot is no different from that of clean dataset. After deploying our OCI and TRP strategies on the one-shot paradigm, the attack performance significantly improve, even better than full-shot in tests where only the first frame or the first 5 frames are poisoned.

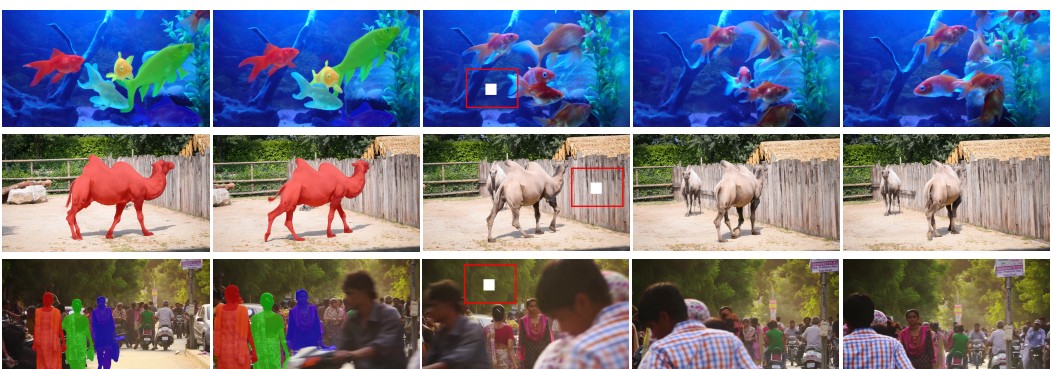

Figure 11: More Visualizations of AOT on the DAVIS 2017 and YouTube-VOS 2019 datasets.

## A.6 RESISTANCE TO MODEL PRUNING

We also implement the popular pruning defense, which is a method of eliminating a backdoor by removing dormant neurons for clean inputs. We first test the backdoored AOT model with 10% clean images from the training set to determine the average activation level of each neuron in the last convolutional layer. Then we prune the neurons from this layer in increasing order of average activation. As shown in Fig. 10, the performance on the poisoned test even decreases (instead of increasing) with the increase in the pruning rate. The performance on clean test also decreases due to pruning. These results demonstrate that our method is resistant to model pruning.

## A.7 MORE VISUALIZATIONS

As shown in Fig. 11, we present more visualization results of backdoored AOT model on DAVIS 2017 and YouTube-VOS 2019 datasets. It can be seen that the model initially correctly segment the specified objects, but when the trigger appeared, all objects in subsequent frames failed to segment.

