# OpenReview forum: "One Shot, One Kill: Attacking Video Object Segmentation with a Single Frame"
_ICLR.cc/2026/Conference — ICLR 2026 Conference Withdrawn Submission_

### Official Review · Reviewer_5w3T · 2025-10-30

**Soundness:** 3
**Presentation:** 3
**Contribution:** 3
**Rating:** 4
**Confidence:** 4

**Summary:**

This paper introduces OSBA, the first backdoor attack on video object segmentation. By injecting a trigger into a single frame during training, the attack causes persistent segmentation failures in subsequent frames at inference. To overcome the transient nature of single-frame triggers, the authors propose two strategies: Object-Centroid Implantation (OCI) and Trigger-Region Perturbation (TRP). Extensive experiments on multiple datasets and models, including digital and physical settings, validate the effectiveness and robustness of the approach.

**Strengths:**

1.This is the first time backdoor threats to VOS in safety-critical scenarios have been revealed, providing a clear and practical risk warning.

2.The method is intuitive and easy to follow.

3.Extensive comparisons across both digital and physical environments demonstrate the robustness and transferability of the attack.

**Weaknesses:**

1.In the introduction, the authors state that “the performance of one-shot attack on poisoned test remains unchanged compared to benign test, indicating that backdoor attack has failed.” It should be clarified whether this “failure” refers to the baseline implementation or to the one-shot paradigm itself.

2.Backdoor attack methods for image segmentation already exist [1,2]. Since attacks against image models can be transferred to video models, and the method proposed in this paper is also one-shot, the authors should compare with improved image backdoor methods to demonstrate the independent contribution of OSBA in the "video + single frame" scenario, or clearly explain why such comparison is omitted.

3.In Table 1, DeAOT’s J&F drops to 9.7% on the poisoned test, much lower than AOT. The authors should explain why a structurally stronger model appears more fragile, and provide analysis on architectural vulnerabilities.

4.In OCI, the trigger is placed at the object centroid. However, when a sequence contains multiple targets, it is unclear how the centroid is determined and where the trigger is placed. The paper should clarify this implementation detail and its impact on attack performance.

5.The paper reports results with a 1% poisoning ratio but does not analyze how varying poisoning levels (<1%, 1–5%, >5%) affect attack strength and stealth. It also remains unclear whether there exists an optimal poisoning range. A systematic study of poisoning ratios would strengthen the evaluation.

6.In Section 4.5 Qualitative Results, key experimental details such as shooting distance, lighting conditions, and camera angles are missing. These factors significantly affect reproducibility and should be documented.

References

[1] Haoheng Lan, et al. "Influencer backdoor attack on semantic segmentation." ICLR 2024.

[2] Yiming Li, et al"Hidden backdoor attack against semantic segmentation models." arXiv 2021.

**Questions:**

See Weaknesses

---

### Official Review · Reviewer_8pfz · 2025-10-30

**Soundness:** 2
**Presentation:** 3
**Contribution:** 2
**Rating:** 4
**Confidence:** 4

**Summary:**

This paper proposed a backdoor attack method on video object segmenation task based on one shot trigger. There are two main designs: one is Object-Centroid Implantation (OCI) which poison image on the centric regions of victim object, another is Trigger-Region Perturbation (TRP) which improves generalization capability of trigger at arbitary locations during inference. Experiment shows that the proposed algorithm can achieve effective one-shot attack for VOS task with only 1% training data poisoning on a set of public benchmarks including DAVIS 2017 and YouTube-VOS 2019.

**Strengths:**

1. Compared with multi-frame or full-shot schemes, the proposed "one-shot" attack mechanism is a more practical solution
2. The achieved experiment result is good for existing AOT & DeAOT VOS models, and the ablation study clearly shows the effectiveness of the proposed OCI and TRP module
3. The paper is well-structured and easy to follow

**Weaknesses:**

1. The paper only deals with AOT & DeAOT VOS models which is relatively old, it is not clear whether the proposed mechanism is effective on more recent VOS algorithms such as SAM based Video Object Segmentation?
2. The proposed trigger patterns is not very stealthy, actually, people can easily find the proposed 4 trigger patterns @ Figure 5 in video frames.

**Questions:**

1. It is not clear how the qualitative result in sec 4.5 on physical-world attack is achieved, if the white trigger patch is physically in the scene, then the trigger pattern will always be in the camera frame, right? then the attack may not be the 'one-shot' attack?

**Details Of Ethics Concerns:**

The paper has an ethics statement. But due to the nature of backdoor attacks and their potential for malicious use in real scenarios such as autonomous navigation, video surveillance, and human-computer interaction, a more deep ethics review may be required.

---

### Official Review · Reviewer_7MUF · 2025-11-01

**Soundness:** 2
**Presentation:** 3
**Contribution:** 2
**Rating:** 2
**Confidence:** 4

**Summary:**

This paper proposes OSBA, the first one-shot backdoor on VOS that plants a trigger in a single frame to induce persistent segmentation failure in all subsequent frames, and introduces Object-Centroid Implantation (OCI) and Trigger-Region Perturbation (TRP) to overcome trigger transientness.

**Strengths:**

1. The paper’s structure is clear and easy to understand.
2. The authors provide a case study of a physics experiment.

**Weaknesses:**

1. The chosen VOS model is outdated. The authors not only selected an older model but also a lightweight backbone. Based on experience from other domains, such models tend to be less robust. The authors should use more advanced ViT-based models to validate the effectiveness of their method, for example, SAM 2.
2. The proposed method is overly simple and seems to suggest that this task is easy to attack, so the contribution appears limited.
3. The experimental analysis is too superficial and does not provide interesting findings—for instance, which architectural choices improve performance while reducing robustness. As a result, the paper offers little substantive guidance for designing more robust models in this area.
4. The discussion of related work is insufficient. It lacks a dedicated section on current robustness research in VOS. Although backdoor attacks might be the first in this specific context, there is already substantial work on other types of robustness, e.g., [1][2][3]，etc.

[1] Adversarial attacks on video object segmentation with hard region discovery, TCSVT2023
[2] Exploring the Adversarial Robustness of Video Object Segmentation via One-shot Adversarial Attacks，ACM MM2023
[3] Vanish into Thin Air: Cross-prompt Universal Adversarial Attacks for SAM2, arxiv2025.

**Questions:**

Are there any findings—for example, which architectures improve performance but reduce robustness—and how should VOS models be designed to defend against backdoor attacks?

---

### Official Review · Reviewer_WY8c · 2025-11-01

**Soundness:** 3
**Presentation:** 3
**Contribution:** 2
**Rating:** 4
**Confidence:** 4

**Summary:**

This paper is the first to explore backdoor attacks on VOS. The authors propose a baseline and further explore the One-Shot Back-door Attack (OSBA). Based on OCI and TRP, experiments validate the effectiveness of the method; however, the experimental models are limited, and the conclusions may not have generalizability.

**Strengths:**

1. Exploring backdoor attacks on VOS is beneficial for the secure deployment of the model.

2. The method is intuitive and simple.

**Weaknesses:**

1. In OCI, are the positions of the triggers independent for each frame or is a centroid position optimized across the entire dataset?

2. How are OCI and TRP used in combination? Are they mixed in the data or superimposed on a single image?

3. The network architectures of AOT and DeAoT are basically the same, which means that the generalization ability of the proposed method has not been verified. Experiments should be supplemented with more state-of-the-art models, especially Xmem [1] and Transformer-based OnsVOS [2]. In addition, using smaller models in the experiments may improve the effectiveness of the attack.

4. This paper does not clearly explain the use cases in which this backdoor attack would exist, lacking real-world scenarios. Furthermore, this paper does not discuss the security risks associated with VOS, such as [3].

[1] Xmem: Long-term video object segmentation with an atkinson-shiffrin memory model, ECCV 2022

[2] OneVOS: Unifying Video Object Segmentation with All-in-One Transformer Framework, ECCV 2024

[3] Exploring the Adversarial Robustness of Video Object Segmentation via One-shot Adversarial Attacks, ACMMM 2023

**Questions:**

See Weaknesses.

---

### Note · Authors · 2025-11-12

I have read and agree with the venue's withdrawal policy on behalf of myself and my co-authors.